# Preliminary Results on the Effects of Soybean Isoflavones on Growth Performance and Ruminal Microbiota in Fattening Goats

**DOI:** 10.3390/ani14081188

**Published:** 2024-04-15

**Authors:** Yuexin Shao, Junhong Xu, Mengyu Wang, Yalun Ren, Manhong Wei, Bowen Tian, Jun Luo, Juan J. Loor, Huaiping Shi

**Affiliations:** 1College of Animal Science and Technology, Northwest A&F University, Yangling 712100, China; shaoyuexin0423@163.com (Y.S.); wmy914624@163.com (M.W.); yalunren2024@163.com (Y.R.); tianbowen310@163.com (B.T.);; 2Weinan Agricultural Products Quality and Safety Inspection and Testing Center, Weinan 714000, China; xujunhong000@soho.com; 3College of Animal Engineering, Yangling Vocational & Technical College, Yangling 712100, China; nwsuafwmh@163.com; 4Department of Animal Sciences and Division of Nutritional Sciences, University of Illinois, Urbana, IL 61801, USA

**Keywords:** soybean isoflavones, fattening goats, ruminal microflora, growth performance

## Abstract

**Simple Summary:**

In this study, we assessed the impact of soybean isoflavones (SIFs) on the health of fattening castrated goats, considering growth performance, slaughter performance, serum parameters, meat quality, and ruminal microbiota. The results indicated that oral supplementation with 100 mg/d of SIFs led to changes in growth performance and non-carcass components. This study also demonstrated the beneficial effect of oral supplementation with 100 mg/d of SIFs on ruminal flora composition. Our results provided a scientific basis for SIFs to improve growth performance.

**Abstract:**

Soybean isoflavones (SIFs), a group of secondary metabolites, have antioxidant, anti-inflammatory, and hormone-like activities. Supplementation with SIFs in the diet was reported to promote lactation performance in ruminants. The present study was performed to further decipher the effect of various concentrations of SIFs on growth and slaughter performance, serum parameters, meat quality, and ruminal microbiota in fattening goats. After a two-week acclimation, a total of 27 5-month-old Guanzhong male goats (18.29 ± 0.44 kg) were randomly assigned to control (NC), 100 mg/d SIF (SIF1), or 200 mg/d SIF (SIF2) groups. The experimental period lasted 56 days. The weight of the large intestine was greater (*p* < 0.05) in the SIF1 and SIF2 groups compared with the NC group. Meat quality parameters indicated that SIF1 supplementation led to lower (*p* < 0.05) cooking loss and shear force (0.05 < *p* < 0.10). The 16S rRNA sequencing analysis demonstrated that SIF1 supplementation led to lower (*p* < 0.05) proportions of *Papillibacter* and *Prevotellaceae_UCG-004* but greater (*p* < 0.05) *CAG-352* abundance in the rumen; these responses might have contributed to the improvement in production performance. In conclusion, meat quality and ruminal microbiome could be manipulated in a positive way by oral supplementation with 100 mg/d of SIFs in fattening goats. Thus, this study provides new insights and practical evidence for the introduction of SIFs as a novel additive in goat husbandry.

## 1. Introduction

Breeding of dairy goats is an important focus in many developing countries and generates a relatively stable income in areas where these livestock species are raised, such as in *Shaanxi* Province in China [1,2]. Young dairy goats possess several outstanding characteristics including liveliness, adaptability, and strong survival ability [3]. From a human nutrition standpoint, meat from fattening goats has a desirable fatty acid profile [4]. Currently, barn-feeding is universal in most large-scale dairy goat farms. In this type of management, male kids primarily obtain nourishment from the basal diet, which helps them develop immunity and promotes healthy growth. Previous studies have indicated that dietary intake and nutrient digestibility of dairy goats increase with age, but the basal diet alone is insufficient to meet their nutritional needs [5,6]. Thus, it is necessary to find innovative approaches to optimize production during the fattening period.

Numerous studies have demonstrated that phytoestrogens can benefit the growth performance and intestinal health of livestock [7,8,9]. Among the phytoestrogens, isoflavones extracted from bean products (SIFs) are considered natural plant estrogens [10] that can interact with endogenous estrogens in the animal thereby enhancing growth and improving feed conversion efficiency [11]. In the longissimus muscle of Chinese mini-pigs, SIFs enhanced fat deposition and modulated mRNA abundance of myokines and lipogenic genes, underscoring its potent biological activities [12]. The fact that high intake of SIFs from legumes had a negative impact on animal performance underscored the need to evaluate optimal doses of exogenous SIFs [13,14].

Ruminal microorganisms play a crucial role in various processes such as digestion, nutrient absorption, pathogen resistance, and stress response in ruminants [15,16]. In the context of phytoestrogens, it is worth noting that SIFs could be metabolized by gastrointestinal microorganisms, which impacts their bioavailability and bioactivity. An in vivo study revealed that SIFs and their metabolite e-quol produced by ruminal microorganisms decreased operational taxonomic units (OTUs) and reduced microbial richness in cows [8]. Whether SIF supplementation regulates ruminal microbiota in fattening goats is unknown.

We hypothesized that SIFs would result in an improvement in production outcomes in fattening goats. Thus, in the current study, we intended to investigate the effect of SIFs on growth performance, slaughter performance, serum parameters, and meat quality. Moreover, the 16S rRNA sequencing method was employed to further reveal the regulatory effect of SIFs on ruminal microbiota.

## 2. Materials and Methods

### 2.1. Experimental Design and Animal Management

After a 2-week adaption period, a total of 27 five-month-old healthy Guanzhong dairy goats (18.29 ± 0.44 kg) were randomly allotted into three treatments with nine animals per group. This study lasted for 8 weeks, with goats being acclimated to the experimental facility and diet for the first 2 weeks. The basal diet with no SIFs (NC) and the treatment groups were supplemented orally with 100 mg/d of SIFs (SIF1) or 200 mg/d of SIFs (SIF2). The doses of SIFs used in this study were identified following our preliminary experiment and an earlier report [17]. Twenty-seven Guanzhong dairy goats were raised in fenced areas, with three goats allocated per fenced area, in a controlled temperature at 26 and 32 °C with a 12 h light/dark cycle. All goats had free access to water and trace-mineralized salt blocks. Before feeding roughage, SIFs were thoroughly commixed with concentrate feed by hand and fed to goats to ensure complete expenditure. The basal diet was formulated based on the National Research Council (2012) nutrient requirements, and the composition and nutrient levels of the diet are reported in the Appendix A. The soybean isoflavone (yellow powder, purity > 80%) product used in this work was obtained from Xi’an Ci Yuan Biotechnology Co., Ltd. (Xi’an, China) (Appendix A). Diets were supplied to the goats twice daily at 07:00 and 15:00 h.

### 2.2. Growth and Feed Intake Data

When the trial began, feed intake was measured and recorded daily. Goats were separately weighed on day 0 and were then weighed every two weeks at the same time in the morning. The average daily gain (ADG) was determined on the basis of the results of feed intake and body weight (BW) on days 0, 14, 28, 42, and 56. During the feeding experiment, the daily feed intake (as-fed) of goats in each pen was recorded, including the offered and refused feed. The average daily feed intake (ADFI) was calculated according to the following formula: ADFI = feed offered-feed refused. The feed conversion ratio (FCR) was determined as ADG/ADFI during the experimental period.

### 2.3. Body Measurements

Body measurements (BMs) were recorded every two weeks. Relevant growth traits including head length, withers height (WH), rump height (RH), body length (BL), chest girth (CG), chest depth (CD), cannon bone circumference (CBC), and abdominal girth (AG) were also recorded. The specific body dimensions measured are depicted in the Appendix A [18]. During the data collection process, a total of four people were involved. They were divided into different groups, with two people responsible for taking measurements and two people responsible for recording the collected data. The measurements were focused on various growth traits, with one person measuring head length, WH, RH, and BL, while the other person measured CG, CD, CBC, and AG.

### 2.4. Slaughter Procedure, Carcass and Non-Carcass Morphometric Measurements, and Butchering

On the 56th day, three goats with the closest average BW in each group were selected and killed at a slaughterhouse using standard commercial procedures by personnel from the College of Animal Science and Technology of Northwest A&F University who hold the certificate of laboratory animal practitioners. During the slaughter process, animals were stunned in the atlanto-occipital region, and then blood was collected through carotid and jugular vessels. After skinning and evisceration, the head (specifically at the atlanto-occipital joint) and extremities (specifically at the metacarpal and metatarsal joints) of each goat were acquired. Non-carcass components such as head, four feet, pelt, heart, spleen, kidneys, fat, and emptied and cleaned digestive tract (rumen, reticulum, omasum, abomasum, small intestine, and large intestine) were weighed as previously described [19]. The length of the large and small intestine was measured after placing it on a flat surface. Morphometric measurements such as carcass weight, carcass depth, GR value, net meat weight, bone weight, ribeye muscle area, dressing percentage, neat percentage, carcass neat percentage, and meat–bone ratio were measured. Specific measurements included carcass weight (after removing the head, feet, haslet, and skin, then preserving at 4 °C for 24 h), chest depth (the distance between sternum and withers), carcass length (the distance between anterior edge of the pubic bone and anterior edge of the first rib at its midpoint), GR value (a dedicated measuring pen was used to measure tissue thickness vertically on the surface of the tissue between the 12th and 13th ribs, 11 cm from the midline of the spine), and ribeye muscle area (the cross-sectional area of the LD muscle between the 12th and 13th ribs of the carcass; the outline of the cross-section of the ribeye muscle was drawn with sulfuric acid drawing paper). Dressing percentage, neat percentage, and carcass neat percentage were calculated, respectively, using the following methods.
Dressing percentage = carcass weight/slaughter body weight
Net meat percentage = net meat weight/slaughter body weight
Carcass net percentage = net meat weight/carcass weight.

### 2.5. Meat Quality Parameters

Water loss and cooking loss were measured to assess the physical characteristics of meat. A longissimus dorsi muscle sample was cut into 3 cm × 2 cm × 1 cm pieces of flesh. The meat was placed in an inflatable plastic bag, ensuring no contact between the sample and the bag. After suspending at 4 °C for 24 h, the meat was removed from the bag and any surface moisture was gently wiped off using filter paper. Then, the meat weight was measured and water loss was calculated by expressing the percentage of weight lost by meat as a percentage of the initial weight. Approximately 30 g of longissimus dorsi muscle sample was placed in a sealed plastic bag and boiled in a water bath at 80 °C for 45 min. After boiling, the sample was cooled to room temperature, any surface moisture was gently wiped off using filter paper, and the weight was measured again. The cooking loss was calculated by expressing the percentage of weight lost by the sample compared with the initial weight. The cooking loss meat samples were tested for shear force by placing them in an ice box at 4 °C for 3 h. The shear force (N) was measured using the SM-8007 meat shear force measuring instrument with a dovetail blade (thickness of 3 mm, internal blade angle of 60°, and internal angle incision height of 4 mm) and a corresponding anvil bed (anvil bed mouth width of 4 mm). The shear force measuring instrument had a shear speed of 10 mm/s. The samples were placed on the anvil bed of the instrument, and the knife edge was positioned perpendicular to the muscle fiber direction to cut the meat sample. The maximum shear value during cutting was recorded, and the data measured were averaged.

### 2.6. Blood Parameters

After overnight fasting, blood was collected via the jugular vein at 6 am on weeks 0, 4, and 8. Blood samples obtained from each goat were placed into a vacuette tube with lithium heparin (Greiner Bio-One GmbH, Kremsmünster, Austria) and then centrifuged at 3500× *g* for 5 min at 4 °C to collect serum, which was stored at −80 °C for further analysis. The levels of blood urea nitrogen (BUN), total cholesterol (TC), total triglyceride (TAG), and total protein (TP) were determined using a Biochemical Analytical Instrument (Beckman CX4). The following growth hormone (GH) and SIF commercial ELISA kits were used for hormone determination according to the manufacturer’s instructions: growth hormone (MB-4795A, Jiangsu Meibiao Biotechnology Co., Ltd., Nanjing, China) and SIFs (FT-P9S1826X, Jiangsu Meibiao Biotechnology Co., Ltd., Nanjing, China). All analyses were conducted on the same day and included a standard curve.

### 2.7. Ruminal Microbiota Sampling and Microbial Diversity Analyses

Ruminal fluid samples were collected from 3 healthy goats with the closest average body weight in each group. Using an oral stomach tube (MDW16, Chengdu Huazhi Kaiwu Co., Ltd., Chengdu, China), the samples were collected through the oral cavity following the method described by a previous study [20]. The collection was carried out at 4 h after feeding on the morning of the 57th day. The first 50 mL of ruminal fluid was discarded and then the filtered ruminal fluid was collected after squeezing through a 4-layer sterile gauze. The samples of ruminal digesta were shipped to Beijing Novogene Technology Co., Ltd. (Beijing, China) for 16S rRNA gene sequencing. According to the manufacturer’s instructions, the genomic DNA of ruminal fluid samples was extracted through CTAB assays as described by Villegas-Rivera et al. [21]. The library was constructed using the TruSeq^®^ DNA PCR-Free Sample Preparation Kit and subsequently quantified based on Qubit and Q-PCR. After library construction, sequencing was performed using the NovaSeq6000 system. Effective clustering tags for all samples were obtained using the Uparse algorithm (Uparse v7.0.1001, http://www.drive5.com/uparse/, (accessed on 5 May 2022)). The sequences were clustered into multiple operational taxonomic units (OTUs) with 97% identity, and then representative OTU sequences were chosen. Species annotation of OTU sequences was performed using Mothur and SILVA138 (http://www.arb-silva.de/, (accessed on 5 January 2023)). R software (Version 2.15.3) was used to draw dilution curves and conduct difference analysis between groups according to Alpha diversity indices. Qiime software (Version 1.9.1) was employed to identify Observed-OTUs, Shannon, and Goods-coverage indexes.

### 2.8. Statistical Analysis

Effects of SIFs on the BW, ADG, ADFI, FCR, biometric measurements before slaughter, and serum biochemical variables were analyzed using SPSS 18.0 (IBM SPSS Statistics, SPSS Inc., Chicago, IL, USA). The statistical models for growth performance (BW, ADG, ADFI, and FCR), biometric measurements before slaughter, and serum biochemical variables data included the fixed effects of treatment, week, treatment × week interaction, and the covariate measurement. Post hoc Dunnett’s or Tukey’s test was performed when significant differences were observed. Carcass morphometric measurements, non-carcass morphometric measurements, and meat quality parameters were analyzed using one-way ANOVA via the generalized linear model procedure in SPSS 18.0. Duncan’s multiple range test was performed to analyze the differences between means. Least squares means are reported and treatment effects were considered to be significant at *p* < 0.05. Trends were reported at 0.05 < *p* < 0.10.

## 3. Results

### 3.1. Growth Performance

In growth performance, no interaction (*p* > 0.1) was found between dietary treatment and time (Table 1). SIF supplementation did not exhibit any significant effect on ADFI (T × W; *p* = 0.511). Furthermore, SIF supplementation did not exhibit any significant effect on ADG (T × W; *p* = 0.147). In addition, SIF supplementation demonstrated an impact on FCR (T × W; *p* = 0.083) after the 8-week treatment. Goats fed SIF1 had enhanced FCR during days 14 to 28, but goats fed SIF2 had decreased FCR during days 42 to 56.

### 3.2. Biometrics and Morphometrics

As shown in Table 2, there was an interaction (*p* < 0.05) between dietary treatment and time in terms of WH. Goats treated with SIF2 had a significantly lower (*p* < 0.05) WH on day 56 than those in the NC and SIF1 groups. Compared with the NC group, supplementation with SIFs reduced (0.05 < *p* < 0.10, Table 3) eye muscle area in goats. However, there was no significant impact (*p* > 0.1) of SIFs on head length, RH, BL, CG, CD, CBC, slaughter body weight, carcass weight, carcass depth, carcass cannon bone circumference, breast muscle thickness, GR value, net meat weight, bone weight, dressing percentage, neat percentage, carcass neat percentage, and meat–bone ratio.

### 3.3. Mass of Non-Carcass Components

Compared with the goats only fed a basal diet, SIF supplementation led to significantly lower (*p* < 0.05) lung weight, but elevated (*p* < 0.05) kidney weight at the end of the 8-week trial (Table 4). Furthermore, the weight of the large intestine was significantly greater (*p* < 0.05) after feeding with SIFs. Nevertheless, SIFs did not alter other non-carcass components including head, four feet, pelt, heart, liver, spleen, fat, rumen, reticulum, omasum, abomasum, or small intestine weight. Additionally, SIFs led to a tendency for greater (0.05 < *p* < 0.10) length of the large intestine, but had no obvious influence (*p* > 0.1) on small intestine length.

### 3.4. Meat Quality

There was no obvious difference (*p* > 0.1) in the water loss rate among NC, SIF1, and SIF2 groups (Table 5). However, in comparison with the goats fed the basal diet, shear force tended to be lower (0.05 < *p* < 0.10) in goats fed SIF1. In addition, cooking loss was significantly lower (*p* < 0.05) after goats were fed SIF1.

### 3.5. Biochemical Indicators in Serum

As shown in Table 6, there was an interaction (*p* < 0.05) between dietary treatment and time in terms of serum SIF levels. The SIF levels in the SIF1 group were significantly greater (*p* < 0.05) than in the NC and SIF2 groups, while no significant difference (*p* > 0.1) was detected between the NC and SIF2 groups. Furthermore, neither SIF1 nor SIF2 had any effect (*p* > 0.1) on TP, TC, TAG, or BUN content in serum.

### 3.6. Ruminal Microbiota Diversity

The increase in number of sequences tended to smooth the rarefaction curve, suggesting that the amount of sequencing data was enough to reflect the large majority of microbiota abundance and diversity in ruminal samples (Figure 1A). Based on 97% identity, the Venn diagram exhibited 860 common OTUs and 81, 63, and 65 unique OTUs in the NC, SIF1, and SIF2 groups, respectively (Figure 1B; Appendix A). In addition, Observed_species and Shannon index were used to analyze α-diversity in ruminal microbiota. The results revealed that SIF2 treatments led to significantly lower (*p* < 0.05) Observed_species and Shannon index. Notably, goats supplemented with SIF2 had a lower (*p* < 0.05) Shannon index than that of the SIF1-induced goats (Figure 1C; Appendix A). The PCOA results indicated that the microbiota structure in the SIF1 and SIF2 groups was different compared with the NC group (Figure 1D).

### 3.7. Ruminal Microbiota Composition

As substantiated in Figure 2A and in the Appendix A, *Bacteroidota*, *Firmicutes*, and *Proteobacteria* were predominant at the phylum level, accounting for nearly 90% of the phyla. Goats supplemented with SIF2 had a greater relative abundance of *Bacteroidota* than that of the NC and SIF1 groups. Furthermore, SIF treatment led to a lower relative abundance of *Firmicutes* and the SIF2 group had a more obvious effect than the SIF1 group. The relative abundance of *Proteobacteria* was also greater in response to SIFs, with SIF2 displaying a stronger effect. At the genus level, relative to the goats in the NC group, consumption of SIF1 led to significantly lower proportions of *Papillibacter* and *Prevotellaceae_UCG-004*, but greater (*p* < 0.05) *CAG-352* abundance. In addition, goats treated with SIF2 exhibited lower (*p* < 0.05) relative abundance of *Desulfovibrio*, *NK4A214_group*, *Ruminococcus*, *Alloprevotella*, and *Lachnospiraceae_NK4A136_group* than that of the NC group (Figure 2B; Appendix A).

As reported in Figure 3A and in the Appendix A, among three diets, the majority of sequences were grouped into Transporters, General function prediction only, and DNA repair and recombination proteins. Furthermore, compared with the NC group, in the SIF1 group, the sequences were significantly enriched in tropane and piperidine while pyridine alkaloid biosynthesis was lower (Figure 3B). In goats fed SIF2, the predicted main functional terms in the ruminal microbiota communities were associated with homologous recombination, valine, leucine and isoleucine, biosynthesis, C5-branched dibasic acid metabolism, biotin metabolism, tryptophan metabolism, lysosome, and cell division (Figure 3C).

## 4. Discussion

Soybean isoflavones are known for their estrogenic or antagonistic properties, with most of their physiological effects being mediated by estrogen receptors (ERs) such as ERa and ERb [22]. In this study, it was found that high concentrations of SIFs had inhibitory effects. It is hypothesized that at higher concentrations, SIFs may interact with the ligand-binding domain of ERb, exhibiting an anti-estrogenic effect [23]. Few studies have investigated the influence of SIFs on growth performance, carcass morphometric measurements, and non-carcass components in goats. For example, feeding a diet supplemented with the isoflavone biochanin A did not significantly affect dry matter intake or growth performance in mid-lactation Saanen dairy goats [24]. The discrepancy between our results and those of Xu et al.s’ study might be partly ascribed to the different levels of SIFs that were fed and the different physiological stages of the animals used. Considering the few published studies pertaining to the association between SIFs and fattening goats, we also compared our results with those obtained from pigs or other animals. In crossbred piglets, Li et al., demonstrated that SIFs enhanced BW on day 72 of life and led to greater feed intake and villus height-to-crypt depth ratio [25]. Taken together, despite the lack of change in carcass morphometric measurements and body size traits due to feeding SIFs, the better growth performance and mass of the abomasum and large intestine underscored its benefits. Future research is needed to assess whether SIFs enhance performance due to better digestive capacity.

A previous study indicated that feeding a diet with soy waste caused higher kidney weight in post-wean kids [26], which seems to agree with the greater kidney weight in response to feeding SIF1 and SIF2. The reasons for the increase in kidney weight and whether this increase affected excretory functions in the animal are unknown. The ribeye muscle area is an important endpoint during the fattening period and is positively correlated with the percentage of lean meat deposition [27]. The significant decrease in ribeye muscle area observed with supplementation of SIF2 indicates a potential adverse effect of high dietary levels of SIFs on meat quality. The high concentration of SIFs may hinder growth and development by exerting an anti-estrogenic effect. This study provides a comparative analysis of the meat quality properties of SIF supplementation in goats. However, it is important to note some limitations. The small sample size of experimental goats and limited tissue available for sampling hindered a comprehensive evaluation of muscle fiber morphology and muscle bundle properties. Additionally, relying solely on SIF treatment for 8 weeks may not accurately depict the entire process of goat growth. These limitations should be taken into consideration for future studies.

Meat quality is influenced by various parameters such as water-holding capacity and shear force [28]. The rate of cooking loss is typically used to measure muscle hydrodynamics, with reduced cooking loss and shear force being synonymous with higher meat quality, i.e., both indicative of smoother and softer meat. The fact that supplementation with SIF1 led to lower shear force and cooking loss, which enhanced meat quality, agrees with poultry experiments in which SIFs enhanced meat quality [29,30]. Collectively, the present data demonstrated that adding 100 mg/d of SIFs to the diet was effective in improving meat quality in fattening goats.

The ruminal microbiota plays a crucial function in promoting nutrient digestion and utilization, which are closely associated with growth performance and production efficiency [31]. A previous study demonstrated that several species from *Prevotella* play an important role in protein fermentation [32]. Thus, we further investigated the influence of SIFs on ruminal microbiota using 16S rRNA gene sequencing technology. The observation that α-diversity was decreased after SIF1 and SIF2 treatments was inconsistent with a recent study investigating the effect of biochanin A (an isoflavone phytoestrogen) on the ruminal microbiome of dairy goats [24]. We believe that the main reason for these inconsistent results lies in the different growth periods during which the samples were collected and analyzed.

At the phylum level, *Bacteroidota*, *Firmicutes*, and *Proteobacteria* were among the most predominant ruminal microbiota in the present study, an observation that agrees with an earlier investigation [33]. In our study, *Papillbacter* represented only 0.03–0.5% of 16S rRNA gene abundance when the ruminal samples were taken at 57d. At the genus level, dietary SIF1 inhibited the growth and proliferation of *Papillibacter*, which was reported previously to be correlated with decreased feed efficiency in ruminants [34]. Thus, the lower relative abundance of *Papillibacter* might be the main reason for the increasing trend of FCR in goats fed SIF1 and SIF2 throughout the entire experimental period. Although SIF1 led to lower numbers of *Prevotellaceae_UCG-004*, which belongs to the *Prevotellaceae* family, it has been reported that some members of *Prevotellaceae* could benefit from starch degradation in the rumen [35,36]. Thus, the overall benefits of lower *Prevotellaceae_UCG-004* in response to feeding SF1 cannot be fully ascertained in the present study. The greater abundance of *CAG-352* in rat feces was associated with better systemic indices of insulin sensitivity during pregnancy [37]. Data from a cattle experiment indicated that the amount of *CAG-352* in the rumen was positively related to total protein, albumin, and non-esterified fatty acid levels in serum [38]. The fact that the relative abundance of *CAG-352* in the rumen was enhanced when feeding SIF1 suggested it may have a positive impact on the rumen. *Desulfovibrio* is a sulfate-reducing bacteria and a previous study linked the increase in this species with a decrease in lactate concentration in the rumen [39]. Of note, in this study, Desulfovibrio represents <0.5% of the sequences, but other lactate utilizers (Megasphaera and Selenomonas) are even less abundant (Appendix A). Thus, the significant inhibition of *Desulfovibrio* abundance in animals fed SIF2 could promote lactate production in the rumen, which needs to be further investigated in the future. The *NK4A214_group* and *Ruminococcus* are members of the *Ruminoccaceae* family, which are involved in the degradation of starch [40,41]. *Lachnospiraceae_NK4A136_group* was reported to participate in the degradation of complex polysaccharides [42]. In the present investigation, SIF2 inhibited the relative abundance of *NK4A214_group*, *Ruminococcus*, and *Lachnospiraceae_NK4A136_group*, suggesting that it had no beneficial effect on digestion efficiency in the rumen. A greater abundance of *Alloprevotella* was associated with enhanced muscle growth and improved meat quality [43]. In our study, SIF2 decreased the relative abundance of *Alloprevotella*, which might not be a beneficial consequence for meat quality in fattening goats. However, despite the high dose of SIFs (SIF2) showing a potentially negative impact on ruminal microbiota, the low dose of SIFs (SIF1) has promising application prospects in fattening goats. The two concentrations examined in our study were insufficient to fully describe the effects of SIFs on rumen microorganisms in dairy goats. Since SIFs are phytoestrogens, future research should take into account the potential beneficial effects of low concentrations. Together, the data indicated that SIFs fed at a lower dose could be used in fattening goats without risks of negative effects on the microbiota or performance.

## 5. Conclusions

We conducted a preliminary assessment of the impact of SIFs on the performance of fattening goats considering growth and slaughter data, serum parameters, meat quality, and ruminal microbiota. Oral supplementation with 100 mg/d of SIFs improved meat quality with decreased shear force and cooking losses. In addition, feeding 100 mg/d of SIFs led to a lower relative abundance of *Papillibacter* and *Prevotellaceae_UCG-004* and a greater abundance of *CAG-352*, both of which suggested a positive effect. The results provided a scientific basis for the application of SIFs to improve the growth performance of fattening goats. This study demonstrated that 100 mg/d SIF supplementation could benefit meat quality and ruminal microbiota. Thus, SIFs are a potential feed additive for improving productive efficiency in goats.

## Figures and Tables

**Figure 1 animals-14-01188-f001:**
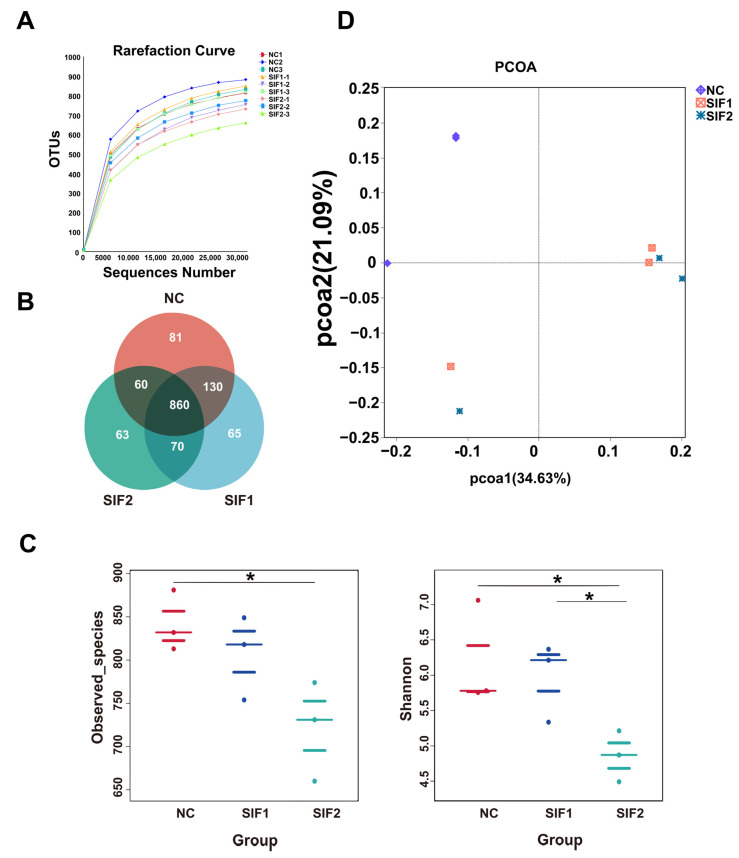
Effects of SIF supplementation on ruminal microbiota diversity in goats. (**A**) Dilution curve; (**B**) Venn diagram; (**C**) α-diversity; and (**D**) PCOA. * *p* < 0.05.

**Figure 2 animals-14-01188-f002:**
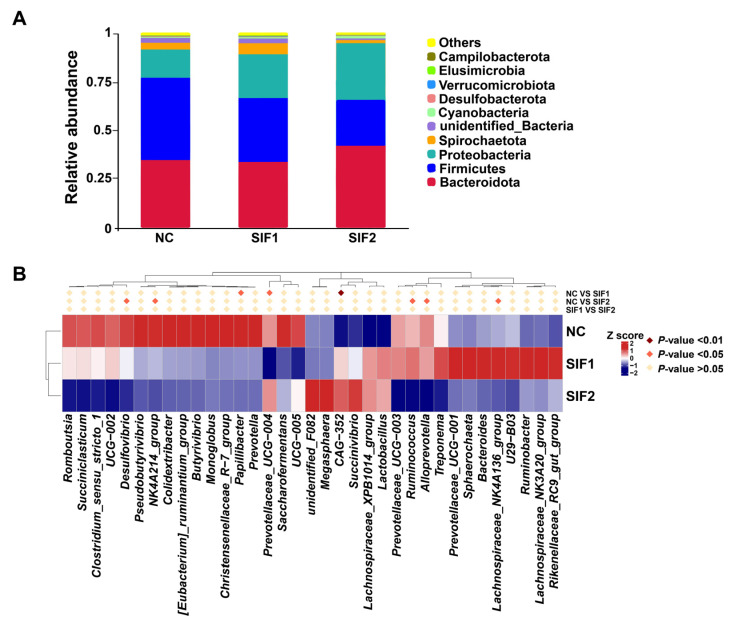
Effects of SIF supplementation on ruminal microbiota composition in goats. (**A**) Microbial composition at the phylum level and (**B**) microbial composition at the genus level.

**Figure 3 animals-14-01188-f003:**
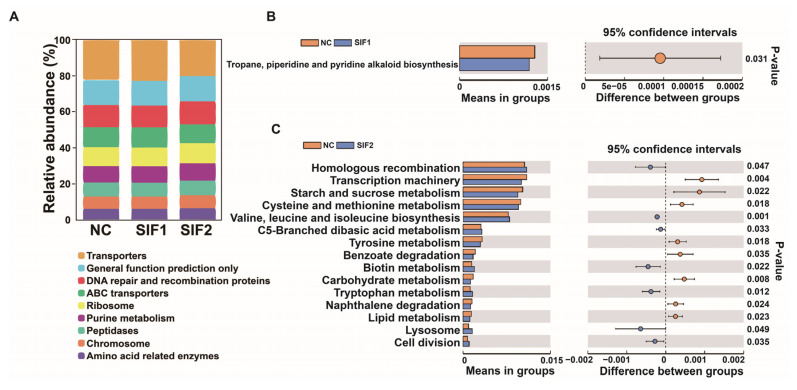
Predictive functional profiling of 16S rRNA sequencing results. (**A**) A bar chart of functional prediction; (**B**) SIF1 vs. NC; and (**C**) SIF2 vs. NC.

**Table 1 animals-14-01188-t001:** Effect of SIF supplementation on growth performance.

Time	Control	SIF1	SIF2	SEM	*p*-Value		
					Diet	Time	Diet × Time
BW, kg							
Day 0	18.32	18.38	18.16	0.35	0.976	<0.01	0.149
Day 14	20.34	20.64	20.06				
Day 28	22.03	21.70	21.83				
Day 42	24.84	25.24	24.71				
Day 56	26.78	26.99	26.83				
ADG, g							
Days 0 to 14	44.84	161.11	163.77	6.17	0.085	<0.01	0.147
Days 14 to 28	138.77	105.95	157.65				
Days 28 to 42	201.19	253.17	205.16				
Days 42 to 56	138.49	133.48	177.14				
Days 0 to 56	151.19	153.67	154.88				
ADFI, kg							
Days 0 to 14	1.05	1.07	1.03	0.09	0.012	<0.01	0.511
Days 14 to 28	1.12	1.14	1.11				
Days 28 to 42	1.25	1.28	1.22				
Days 42 to 56	1.19	1.22	1.18				
Days 0 to 56	1.02	1.11	1.13				
FCR							
Days 0 to 14	6.81	6.78	7.20	0.66	0.004	0.002	0.083
Days 14 to 28	8.42 ^b^	10.56 ^a^	7.25 ^b^				
Days 28 to 42	6.29	5.46	6.27				
Days 42 to 56	10.47 ^a^	10.07 ^a^	7.79 ^b^				
Days 0 to 56	6.84	7.28	7.65				

ADG: average daily gain; ADFI: average daily feed intake; FCR: feed conversion rate; a,b: values within a row with no common letters differ significantly (*p* < 0.05); Nine replicates per treatment (*n* = 9); total number of animals is 27.

**Table 2 animals-14-01188-t002:** Biometric measurements before slaughter.

Item		Control	SIF1	SIF2	SEM	*p*-Value		
						Diet	Time	Diet × Time
Head length	Day 0	12.47	12.23	12.32	0.14	0.856	<0.01	0.184
	Day 28	13.63	13.93	13.97				
	Day 56	14.68	14.52	14.91				
WH	Day 0	56.40	55.99	55.26	0.56	0.720	<0.01	0.021
	Day 28	59.07	60.42	60.77				
	Day 56	66.71	66.74	64.27				
RH	Day 0	54.90	54.23	54.09	0.38	0.855	<0.01	0.189
	Day 28	57.02	57.36	57.61				
	Day 56	59.30	60.08	61.24				
BL	Day 0	51.24	51.46	50.07	0.50	0.580	<0.01	0.680
	Day 28	56.01	54.71	54.90				
	Day 56	59.72	60.14	58.30				
CG	Day 0	60.33	60.64	60.89	0.44	0.979	<0.01	0.550
	Day 28	64.84	62.99	63.03				
	Day 56	66.72	66.79	66.99				
CD	Day 0	26.09	25.74	25.73	0.13	0.957	<0.01	0.153
	Day 28	27.42	26.82	26.85				
	Day 56	27.50	27.09	27.00				
CBC	Day 0	7.42	7.43	7.23	0.07	0.574	<0.01	0.453
	Day 28	7.76	7.66	7.65				
	Day 56	7.97	7.89	7.74				
AG	Day 0	59.50	61.11	60.08	0.47	0.198	<0.01	0.850
	Day 28	63.88	65.21	62.05				
	Day 56	66.84	67.51	66.22				

WH: withers height; RH: rump height; BL: body length; CD: chest depth; CG: chest girth; CBC: cannon bone circumference; AG: abdominal girth; Biometric measurements before slaughter (*n* = 9).

**Table 3 animals-14-01188-t003:** Carcass morphometric measurements.

Item	Control	SIF1	SIF2	SEM	*p*-Value
Slaughter body weight/kg	24.53	26.47	25.13	0.62	0.486
Carcass weight/kg	12.55	12.53	12.76	0.32	0.960
Carcass length/cm	74.83	70.67	70.00	1.40	0.355
Carcass depth/cm	19.50	20.33	20.17	0.40	0.725
GR value/cm	0.92	1.37	1.03	0.13	0.410
Net meat weight/kg	8.57	9.02	8.76	0.20	0.705
Bone/kg	3.45	2.93	3.43	0.16	0.380
Eye muscle area/cm^2^	13.18	9.66	7.09	1.19	0.089
Dressing percentage/%	51.13	47.40	50.81	0.95	0.227
Neat percentage/%	34.96	34.10	34.87	0.30	0.504
Carcass neat percentage/%	68.37	72.17	68.62	0.93	0.184
Meat–bone ratio/%	2.48	3.23	2.55	0.19	0.223

**Table 4 animals-14-01188-t004:** Mass of non-carcass components.

Time	Control	SIF1	SIF2	SEM	*p*-Value
Head/kg	1.50	1.60	1.50	0.03	0.296
Four feet/kg	0.64	0.65	0.72	0.02	0.055
Pelt/kg	1.43	1.63	1.63	0.05	0.125
Heart/g	124.00	116.17	122.83	2.14	0.308
Liver/g	555.17	625.17	596.00	19.60	0.392
Spleen/g	47.50	46.50	49.00	2.81	0.951
Kidneys/g	99.17 ^b^	112.50 ^a^	117.33 ^a^	3.25	0.027
Fat/g	327.33	389.67	384.50	51.65	0.961
Rumen/g	468.83	522.00	496.33	15.64	0.437
Reticulum/g	91.17	82.00	76.17	4.28	0.405
Omasum/g	80.17	76.83	70.00	3.05	0.439
Abomasum/g	143.83	169.67	177.00	7.58	0.093
Large intestine/g	363.83 ^b^	534.33 ^a^	532.50 ^a^	31.03	0.005
Small intestine/g	675.33	642.67	592.67	25.21	0.461
Large intestine/cm	438.50	599.67	582.67	34.59	0.093
Small intestine/cm	2002.00	2020.33	1966.83	52.78	0.935

a,b: values within a row with no common letters differ significantly (*p* < 0.05).

**Table 5 animals-14-01188-t005:** Meat quality parameters of test goats.

Time	Control	SIF1	SIF2	SEM	*p*-Value
Water loss/%	10.81	5.91	6.86	1.17	0.167
Shear force (N)	58.84	44.64	50.62	2.67	0.068
Cooking loss/%	42.19 ^a^	31.87 ^b^	39.39 ^ab^	1.92	0.045

a,b: values within a row with no common letters differ significantly (*p* < 0.05).

**Table 6 animals-14-01188-t006:** Serum biochemical indicators of test goats.

Time	Control	SIF1	SIF2	SEM	*p*-Value		
					Diet	Time	Diet × Time
SIFs/(pg/mL)							
Day 0	29.77	31.29	33.94	1.92	0.133	0.09	<0.01
Day 28	28.78	29.53	24.77				
Day 56	32.63 ^b^	55.74 ^a^	29.28 ^b^				
GH/(μg/L)							
Day 0	19.29	19.45	16.93	0.26	0.156	0.01	0.264
Day 28	18.10	17.07	17.88				
Day 56	16.71	16.98	15.89				
Serum-[TP]/(μg/mL)							
Day 0	342.48	337.97	334.45	7.85	0.368	0.02	0.882
Day 28	295.31	276.23	299.82				
Day 56	296.31	267.70	318.14				
Serum-[TC]/(mmol/L)							
Day 0	2.05	2.07	1.98	0.04	0.724	<0.01	0.119
Day 28	2.25	2.50	2.23				
Day 56	2.14	2.25	2.39				
Serum-[TAG]/(mmol/L)							
Day 0	0.27	0.28	0.29	0.01	0.987	<0.01	0.202
Day 28	0.24	0.26	0.23				
Day 56	0.30	0.28	0.29				
BUN/(mmol/L)							
Day 0	6.63	6.91	6.33	0.25	0.835	<0.01	0.443
Day 28	8.47	8.28	8.63				
Day 56	8.56	8.92	8.08				

GH: growth hormone; TP: total protein; TC: total cholesterol; TAG: total triglyceride; BUN: blood urea nitrogen; a,b: values within a row with no common letters differ significantly (*p* < 0.05).

## Data Availability

The data that support the findings in this study were not deposited in an official repository. These data are available from the authors upon request.

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
