# Peer review of "Preliminary Results on the Effects of Soybean Isoflavones on Growth Performance and Ruminal Microbiota in Fattening Goats"

_animals, 2024, doi:10.3390/ani14081188_

Round 1

Reviewer 1 Report

Comments and Suggestions for Authors

The study investigated different doses of isoflavone on growth performance, carcass traits, ruminal microbiology and blood characteristics. The manuscript is well-prepared. I have following concerns in this paper.

1. Number of replicates for the ruminal microbiology and carcass traits and meat quality evaluation was very low - only three per treatments. Authors are required to justify this low number of replicates.

2. Statistical analysis should use repeated measure model for the repeated variables. Also, I have notices in the tables, that authors have used superscripts when diet or diet x time effect is not significant although in the statistical section they have mentioned that "Post hoc Dunnett’s or Tukey’s test was performed when significant differences were observed." Authors are required to consult with a statistician.

For example in Table 1: diet x time effect was not significant for ADG, but they have used superscripts among diet within a time, which is incorrect beacuse interaction effect is not significant.

For ADFI, main effect of diet was significant, so authors cannot use superscripts among diet within a time. Only they can use against overall mean values of diet.

3. Result section need to rewritten based on the modified results in the tables.

4. In discussion section, authors have not clearly explained why high dose was not better than the low dose although they have recommended for the low dose in the conclusion section.

Author Response

The study investigated different doses of isoflavone on growth performance, carcass traits, ruminal microbiology and blood characteristics. The manuscript is well-prepared. I have following concerns in this paper.

  1. Number of replicates for the ruminal microbiology and carcass traits and meat quality evaluation was very low - only three per treatments. Authors are required to justify this low number of replicates.

Response: Thanks. In this manuscript, we only used 3 biological replicates per group due to limitations in experimental conditions and the small dairy goat industry. As previously published, Thomas et al., also used three replicates to investigate the effect of lactic acid, peroxyacetic acid, a hydrochloric and citric acid blend, and 5% levulinic acid plus 0.5% sodium dodecyl sulfate on goat carcass color from slaughter through 24-h chill, suggesting that three replicates also meet the basic requirements for biological replicates. Similarly, even with fewer repetitions, our data still provided a certain degree of reliability.We deeply appreciative to your suggestion, which provides a direction for our next research. In the future, we will closely pay attention to this issue and take measures to improve the quality of research, such as increasing the number of repetitions and stricter experimental design.

Reference: Thomas CL, Stelzleni AM, Rincon AG, Kumar S, Rigdon M, McKee RW, Thippareddi H. Validation of Antimicrobial Interventions for Reducing Shiga Toxin-Producing Escherichia coli Surrogate Populations during Goat Slaughter and Carcass Chilling. J Food Prot. 2019 Mar;82(3):364-370. doi: 10.4315/0362-028X.JFP-18-298IF: 2.0 Q3 . PMID: 30767673.

  1. Statistical analysis should use repeated measure model for the repeated variables. Also, I have notices in the tables, that authors have used superscripts when diet or diet x time effect is not significant although in the statistical section they have mentioned that "Post hoc Dunnett’s or Tukey’s test was performed when significant differences were observed." Authors are required to consult with a statistician.

For example in Table 1: diet x time effect was not significant for ADG, but they have used superscripts among diet within a time, which is incorrect beacuse interaction effect is not significant.

For ADFI, main effect of diet was significant, so authors cannot use superscripts among diet within a time. Only they can use against overall mean values of diet.

Response: Thanks. According to your suggetation, we have conducted repeated measurement model for the repeated variables and improve tables quality (Table 1 )

  1. Result section need to rewritten based on the modified results in the tables.

Response: Thanks. We have improve the result section based on the modified results in the tables. Line: 216-219

  1. In discussion section, authors have not clearly explained why high dose was not better than the low dose although they have recommended for the low dose in the conclusion section.

Response: Thanks. The discussion were rewritten according to comments.

Soybean isoflavone stands out for its estrogenic or antagonistic attributes. It has been established that most of the physiological effects of isoflavones were mediated by estrogen receptors (ERs) including ERa and ERb. We observed that high concentrations of SIF led to inhibitory effects. We speculate that, at higher concentrations SIF migh ligandbinding domain of ERb and show an anti-estrogenic effect.Line: 312-316.

Reviewer 2 Report

Comments and Suggestions for Authors

The authors examined production metrics, carcass characteristics, serum metabolites and bacterial community composition in meat goats fed a basal diet and the same diet at two levels of soybean isoflavones (SIF). Only a few of the animal metrics displayed even slight differences cs, and only moderate differences were observed in bacterial community composition. Overall, the work appears to have been performed competently, but the reviewer has some issues with methodologies. Specifically, it appears that only a single rumen sample was collected for bacterial community composition analysis from each goat, on day 57, well away from the time period at which the modest effects on growth were observed . Moreover, the sample was collected by stomach tubing, which would have led to a selection for the planktonic community at the expense of the (likely more abundant) particle-associated community. Also, the authors do a lot of comparisons of differences in relative abundance for different taxa, but they do not provide relative abundance data themselves (see Comment to Fig. 2B below). This detracts from any microbiology story that the authors can devise, and it does not help that the authors – in an attempt to relate the microbial community to the few animal differences they did observe – have misstated some of the physiological capabilities of the several of the genera that do show differences in relative abundance with treatment (see Specific Comments below). The authors need to point out some of these limitations in their study, and to explain the microbial data more cogently.

Specific Comments:

L18-19: This opening sentence is unnecessary and can be deleted.

L23: Delete “of fattening castrated goats” (redundant).

L54: Higher than what?

L94: By “in a separate fence”, do the authors mean “in separate pens”?

L97: What is meant by “complete expenditure”? Do the authors mean complete admixture?

L107: Approximately what fraction of the feed was refused? Goats are notoriously capable sorters of feed. How was it assured that the orts were not significantly enriched or depleted in each of the feed components?

L187: By “scraped”, do the authors mean scrapped (i.e., discarded)?

L190-191: More detail needed here on the CTAB method (literature citation, or more detailed explanation of procedure).

L193-197: More detail needed here. Was 2x250 bp sequencing performed, or 2x300 bp sequencing employed? Were the number of sequences normalized to the lowest number of sequences across all samples? Were singletons removed from the analysis? Also, the authors should provide, either here or in the microbiota section of the Results, the range of number of sequences obtained for the different samples. What was the level of Good’s coverage across samples?

Table 1: Indicate units for FCR.

Table 6: Include a footnote defining abbreviations for the serum components.

Fig. 1B: The three replicates for SIF1 and SIF2 are clearly observable, but there appears to be only single point for the NC treatment. Please explain.

L283: Explain the bars and dot symbols in Figure 1.

Fig. 2A, 2C and 3A: Within each of the bar graphs, it is difficult to detect differences by treatment, and there is no indication of which metric significantly differs across treatments. Suggest converting these panels to separate tables, and including them in the Supplementary Information rather than in the main manuscript. This would also allow identifying (via superscripted letters) which treatment means displayed significant differences.

Fig. 2B: The Z-scores do show some major differences in abundance of particular genera. However, because the authors do not provide the relative abundance data themselves, it is difficult for the reader to gauge if any of these differences are likely to contribute to any of the animal metrics observed. For example, there were large differences among treatments in the relative abundance of Papillibacter, but just how abundant was this genus. If it represented (on average) 10% of the bacterial community, these treatment differences may mean something, but if it represents 0.01% of the community, then these treatment differences are likely meaningless. The authors should include an OTU table in the Supplementary files, that provides the relative abundance of each genus within each treatment group.

L320-324: It’s difficult to accept that a greater mass of abomasum and large intestine are necessarily a good thing, if there is not a corresponding improvement in animal growth, performance, or health. Unless digestive function is improved, such mass increases would not seem all that beneficial to the animal, and it would seem to require additional feed resources for biosynthesis and maintenance of this extra mass.

L344-346: This statement is an overinterpretation of the cited reference. Zened et al. do not demonstrate cellulose degradation by any of the Rikenellaceae; instead they only cite Pitta et al., Microb Ecol 59: 511(2010), that a “Rikenella-like” genus “clustered with other genera including Fibrobacter, and suggested that this bacterium is involved in structural carbohydrates degradation.” Association with a cellulolytic genus such as Fibrobacter is hardly evidence that this Rikenella-like genus can degrade cellulose.

L356-360: The problem here is that “feed efficiency” as judged by FCR was only improved by SIF1 during a single two-week period (Table 1). Tying Papillibacter to feed efficiency in this study is not possible because its relative abundance was only measured on d 56, at which time there was no improvement in FCR (Table 1).

L369-373: It is not clear what sort of adverse health effects the authors are considering, but overall they seem unlikely. Firstly, lactate is only problematic at high concentrations (although it was not measured in this study, one would expect lactate not to abundant on the diets used here). And while Desulfovibrio can use lactate, a decrease in its relative abundance would mean little as long as other lactate utilizers were present (such as Megasphaera elsdenii and the highly abundant Selenomonas ruminantium).

L373-378: Several problems here. Firstly, relatively few ruminococci degrade starch (most instead degrade structural polysaccharides such as cellulose and hemicellulose). And regardless, a decrease in their relative abundance means little if other taxa with overlapping degradative capabilities can fill the niche.

L384: Is no doubt what?

Minor edits:

L81: Change “intend” to “intended”.

L88: Change “replicates” to “animals”.

L170: Change “was” to “were”.

L288: Change “population” to “relative abundance”.

L289: Change “amount” to “abundance”.

L334: Delete “The value of”.

L336: Change “synonimus” to “synonymous”.

L373: Change “Ruminococcus family” to “Ruminoccaceae family”

Comments on the Quality of English Language

Minor editing suggested (see Comments and Suggestions for Authors).

Author Response

The authors examined production metrics, carcass characteristics, serum metabolites and bacterial community composition in meat goats fed a basal diet and the same diet at two levels of soybean isoflavones (SIF). Only a few of the animal metrics displayed even slight differences cs, and only moderate differences were observed in bacterial community composition. Overall, the work appears to have been performed competently, but the reviewer has some issues with methodologies. Specifically, it appears that only a single rumen sample was collected for bacterial community composition analysis from each goat, on day 57, well away from the time period at which the modest effects on growth were observed . Moreover, the sample was collected by stomach tubing, which would have led to a selection for the planktonic community at the expense of the (likely more abundant) particle-associated community. Also, the authors do a lot of comparisons of differences in relative abundance for different taxa, but they do not provide relative abundance data themselves (see Comment to Fig. 2B below). This detracts from any microbiology story that the authors can devise, and it does not help that the authors – in an attempt to relate the microbial community to the few animal differences they did observe – have misstated some of the physiological capabilities of the several of the genera that do show differences in relative abundance with treatment (see Specific Comments below). The authors need to point out some of these limitations in their study, and to explain the microbial data more cogently.

 Response: Thank you for your careful review and constructive suggestions regarding our manuscript. We have revised the manuscript in accordance with the comments and marked all the amends on our revised manuscript.

Specific Comments: 

L18-19: This opening sentence is unnecessary and can be deleted.

Response: Thanks for your suggestion. We have deleted this sentence.

L23: Delete “of fattening castrated goats” (redundant).

Response: Thanks for your suggestion. We have deleted it.

L54: Higher than what?

Response: Thanks. We have deleted it.

L94: By “in a separate fence”, do the authors mean “in separate pens”?

Response: Thanks. Actually, twenty-seven Guanzhong dairy goats were raised in fences, three goats per fence in this study. We have corrected this information in the manuscript. Line: 86-88.

L97: What is meant by “complete expenditure”? Do the authors mean complete admixture?

Response: Thanks. “Complete expenditure” means that concentrate feed with SIF supplementation were consumed completely by goats, ensuring that each goat has a certain intake of SIF with 100 mg/d or 200 mg/d. In this study, before feeding roughage to goats, we first mix SIF with concentrate. Due to the dietary characteristics of goats, the SIF-concentrate mixture can be fully consumed, which can ensure a specific intake of SIF. Thank for your suggestion again. We have revised this sentence in the manuscript. Line: 89-90.

L107: Approximately what fraction of the feed was refused? Goats are notoriously capable sorters of feed. How was it assured that the orts were not significantly enriched or depleted in each of the feed components?

Response: Thanks. In this study, more feed was provided to ensure free access for goats, and the remaining feed was collected every day. As you mentioned, goats are notoriously capable sorters of feed, so it is difficult to enable that the orts were not significantly enriched or depleted in each of the feed components. However, in the process of the experiment, we attempted to reduce this error through mixing the feed multiple times during the feeding process of dairy goats. Thanks for your suggestions again. In future feeding experiments, we will pay more attention to this issue and take corresponding measures to decrease experimental errors.

L187: By “scraped”, do the authors mean scrapped (i.e., discarded)?

Response: Thanks for your suggestion. We have modified “scraped” to “discarded” in this manuscript. Line: 183.

L190-191: More detail needed here on the CTAB method (literature citation, or more detailed explanation of procedure).

Response: Thanks for your suggestion. As described by Villegas-Rivera et al., the genomic DNA of ruminal fluid samples was extracted through CTAB assays. We have cited this reference in the manuscript. Line: 186-188.

Reference: Villegas-Rivera G, Vargas-Cabrera Y, González-Silva N, Aguilera-García F, Gutiérrez-Vázquez E, Bravo-Patiño A, Cajero-Juárez M, Baizabal-Aguirre VM, Valdez-Alarcón JJ. Evaluation of DNA extraction methods of rumen microbial populations. World J Microbiol Biotechnol. 2013 Feb;29(2):301-7. doi: 10.1007/s11274-012-1183-2. Epub 2012 Oct 5. PMID: 23054703.

L193-197: More detail needed here. Was 2x250 bp sequencing performed, or 2x300 bp sequencing employed? Were the number of sequences normalized to the lowest number of sequences across all samples? Were singletons removed from the analysis? Also, the authors should provide, either here or in the microbiota section of the Results, the range of number of sequences obtained for the different samples. What was the level of Good’s coverage across samples?

Response: Thanks for your suggestion. The sequences were clustered into multiple operational taxonomic units (OTUs) with 97% identity, and then representative OTUs sequences were chosen. Line: 192-194.

Table 1: Indicate units for FCR.

Response: Thanks. The feed conversion ratio (FCR) was determined as ADG/ADFI during the experimental period. Line: 104.

Table 6: Include a footnote defining abbreviations for the serum components.

Response: Thanks for your suggestion. We have supplemented the footnote in table 6.

Fig. 1B: The three replicates for SIF1 and SIF2 are clearly observable, but there appears to be only single point for the NC treatment. Please explain.

Response: Thanks. We apologize for the poor clarity of the figure 1. We have enhanced the clarity of this figure. Furthermore, as shown in the following figure, there are three points marked in blue for the NC treatment, of which two of the points are tightly adjacent to each other.

Fig. 2A, 2C and 3A: Within each of the bar graphs, it is difficult to detect differences by treatment, and there is no indication of which metric significantly differs across treatments. Suggest converting these panels to separate tables, and including them in the Supplementary Information rather than in the main manuscript. This would also allow identifying (via superscripted letters) which treatment means displayed significant differences.

Response: Thanks. It is sorry for not explaining it clearly in the manuscript. At the phylum level, no significant difference of bacteria was found in this study, thus we only used a bar chart to display these results. Furthermore, figure 3A is a diagram of the functional prediction results, the significant altered predictive functional results were shown in figure 3B and 3C. According to your suggestion, we have supplemented the related tables in the supplementary Table S4 and S6.

Fig. 2B: The Z-scores do show some major differences in abundance of particular genera. However, because the authors do not provide the relative abundance data themselves, it is difficult for the reader to gauge if any of these differences are likely to contribute to any of the animal metrics observed. For example, there were large differences among treatments in the relative abundance of Papillibacter, but just how abundant was this genus. If it represented (on average) 10% of the bacterial community, these treatment differences may mean something, but if it represents 0.01% of the community, then these treatment differences are likely meaningless. The authors should include an OTU table in the Supplementary files, that provides the relative abundance of each genus within each treatment group.

Response: Thanks for your suggestions. We have supplemented the OTUs table and relative abundance table at the genus level in the supplementary Table S3 and S5.

L320-324: It’s difficult to accept that a greater mass of abomasum and large intestine are necessarily a good thing, if there is not a corresponding improvement in animal growth, performance, or health. Unless digestive function is improved, such mass increases would not seem all that beneficial to the animal, and it would seem to require additional feed resources for biosynthesis and maintenance of this extra mass.

Response: Thanks. Future research is needed to assess whether SIF enhances performance due to better digestive capacity.

L344-346: This statement is an overinterpretation of the cited reference. Zened et al. do not demonstrate cellulose degradation by any of the Rikenellaceae; instead they only cite Pitta et al., Microb Ecol 59: 511(2010), that a “Rikenella-like” genus “clustered with other genera including Fibrobacter, and suggested that this bacterium is involved in structural carbohydrates degradation.” Association with a cellulolytic genus such as Fibrobacter is hardly evidence that this Rikenella-like genus can degrade cellulose.

Response: Thanks for your suggestion. We have deleted this reference.

L356-360: The problem here is that “feed efficiency” as judged by FCR was only improved by SIF1 during a single two-week period (Table 1). Tying Papillibacter to feed efficiency in this study is not possible because its relative abundance was only measured on d 56, at which time there was no improvement in FCR (Table 1).

Response: Thanks for your suggestion. According to a previous study, Papillibacter was reported to be correlated with the decreased feed efficiency in ruminants. Furthermore, In the entire experimental period, SIF had a tendency to increase FCR (CON=6.84; SIF1=7.28; SIF3=7.65). Thus, we speculated that this change might be associated with the lower relative abundance of Papillibacter in rumen. We have revised this statement in the latest manuscript. Line: 360-364.

L369-373: It is not clear what sort of adverse health effects the authors are considering, but overall they seem unlikely. Firstly, lactate is only problematic at high concentrations (although it was not measured in this study, one would expect lactate not to abundant on the diets used here). And while Desulfovibrio can use lactate, a decrease in its relative abundance would mean little as long as other lactate utilizers were present (such as Megasphaera elsdenii and the highly abundant Selenomonas ruminantium).

Response: Thanks. We highly approve your suggestion and revised this statement in the latest manuscript. Line: 374-378.

L373-378: Several problems here. Firstly, relatively few ruminococci degrade starch (most instead degrade structural polysaccharides such as cellulose and hemicellulose). And regardless, a decrease in their relative abundance means little if other taxa with overlapping degradative capabilities can fill the niche.

Response: Thanks for your suggestion, which give us a deeper thinking. Although relatively few Ruminococcus family was found in this study, they still play a important effect in starch degradation based on the previous study. Furthermore, no significant difference of other taxa with overlapping degradative capabilities was observed in this study, so we think the decreased relative abundance of NK4A214_group and Ruminococcus caused by SIF2 might have no beneficial effect on starch digestion efficiency in the rumen. We have revised this statement in the latest manuscript. Line: 380-384.

Reference: 1. Gaffney J, Embree J, Gilmore S, Embree M. Ruminococcus bovis sp. nov., a novel species of amylolytic Ruminococcus isolated from the rumen of a dairy cow. Int J Syst Evol Microbiol. 2021 Aug;71(8):004924. doi: 10.1099/ijsem.0.004924. PMID: 34379583; PMCID: PMC8513621.

  1. Zhao FF, Zhang XZ, Zhang Y, Elmhadi M, Qin YY, Sun H, Zhang H, Wang MZ, Wang HR. Tannic Acid-Steeped Corn Grain Modulates in vitro Ruminal Fermentation Pattern and Microbial Metabolic Pathways. Front Vet Sci. 2021 Oct 28;8:698108. doi: 10.3389/fvets.2021.698108. PMID: 34778425; PMCID: PMC8581138.

L384: Is no doubt what?

 Response: Thanks. We have revised this statement. Line: 390-391.

Minor edits:

L81: Change “intend” to “intended”.

 Response: Thanks. We have revised it. Line: 74.

L88: Change “replicates” to “animals”.

 Response: Thanks. We have revised it. Line: 82.

L170: Change “was” to “were”.

 Response: Thanks. We have revised it. Line: 162.

L288: Change “population” to “relative abundance”.

 Response: Thanks. We have revised it. Line: 288.

L289: Change “amount” to “abundance”.

 Response: Thanks. We have revised it. Line: 290.

L334: Delete “The value of”.

 Response: Thanks. We have deleted it.

L336: Change “synonimus” to “synonymous”.

 Response: Thanks. We have revised it. Line: 342.

L373: Change “Ruminococcus family” to “Ruminoccaceae family”

 Response: Thanks. We have revised it. Line: 378-379.

Reviewer 3 Report

Comments and Suggestions for Authors

There is a research gap on dietary supplementation of SIF and its effect on fattening goats.

The experimental design is appropriate. There was no mention whether the operators were blinded to treatments when taken some traits like meat traits. The use of SPSS and the outlined models are adequate.

There were some non-significant results. Could the authors elaborate on that?

SIF led to significantly lower eye muscle area. Why? Could the authors give some explanations? SIF1 and SIF2 treatments led to significantly lower Observed_species and Shannon index. Could the authors also speculate on the reason?

Could the authors suggest directions for future research? Could they also give clear recommendations for the use of SIF in goat diets, including any considerations or precautions that should be taken.

Comments on the Quality of English Language

The English is correct. One typo is noted. (Duplication of word)

Author Response

There is a research gap on dietary supplementation of SIF and its effect on fattening goats.

The experimental design is appropriate. There was no mention whether the operators were blinded to treatments when taken some traits like meat traits. The use of SPSS and the outlined models are adequate.

There were some non-significant results. Could the authors elaborate on that?

Response: Thanks. Despite the lack of change in carcass morphometric measurements and body size traits due to feeding SIF, the better growth performance and mass of abomasum and large intestine underscored its benefits. Future research is needed to assess whether SIF enhances performance due to better digestive capacity. Line 326-330:

SIF led to significantly lower eye muscle area. Why? Could the authors give some explanations?

 Response: Thanks. The discussion were rewritten according to comments. Line: 336-339.

The significant decrease in ribeye muscle area observed with supplementation of SIF2 indicates a potential adverse effect of high dietary levels of SIF on meat quality. The high concentration of SIF may hinder growth and development by exerting an anti-estrogenic effect.

SIF1 and SIF2 treatments led to significantly lower Observed_species and Shannon index. Could the authors also speculate on the reason?

Response: Thanks. Soybean isoflavone is known for its estrogenic or antagonistic properties, with most of its physiological effects being mediated by estrogen receptors (ERs) such as ERa and ERb. In this study, it was found that high concentrations of SIF had inhibitory effects. It is hypothesized that at higher concentrations, SIF may interact with the ligand-binding domain of ERb, exhibiting an anti-estrogenic effect.

Could the authors suggest directions for future research? Could they also give clear recommendations for the use of SIF in goat diets, including any considerations or precautions that should be taken.

Response: Our data indicated that SIF fed at a lower dose could be used in fattening goats without risks of negative effects on the microbiota or performance.

Round 2

Reviewer 1 Report

Comments and Suggestions for Authors

Authors have revised most comments. But the major issue is replicate. In the previous review, it was mentioned that there is a major issue for the number of replicates for some variables. Authors cite a paper indicating that there was three replicates, it is not the scientific justification. Moreover, the quality of that journal was relatively low. Authors should justify the replicates number, the weakness of the design, etc.

Statistical presentation has still issue.

For example, Table 1: interaction effect is not significant, but they use superscripts, for example, for FCR; for ADFI diet effect is significant, but no superscript has been used.

Table 3: Eye muscle area is not significant, but superscripts have been used.

Table 4, 5: Superscripts are not defined properly.

Figure 2 A: Were there no significant effect at the phylum level? It seems Firmicutes in control was greater.

Author Response

Authors have revised most comments. But the major issue is replicate. In the previous review, it was mentioned that there is a major issue for the number of replicates for some variables. Authors cite a paper indicating that there was three replicates, it is not the scientific justification. Moreover, the quality of that journal was relatively low. Authors should justify the replicates number, the weakness of the design, etc.

Response: Thanks. In this manuscript, we only used 3 biological replicates per group due to limitations in experimental conditions and the small dairy goat industry. As previously published, Deng et al., also used three replicates to investigate the comparison of muscle fiber characteristics and meat quality between newborn and adult Haimen goats. This study provides a comparative analysis of the meat quality properties of SIF supplementation goat. However, it is important to note some limitations. The small sample size of experimental goats and limited tissue available for sampling hindered a comprehensive evaluation of muscle fiber morphology and muscle bundle properties. Additionally, relying solely on SIF treatment for 8 weeks may not accurately depict the entire process of goat growth. These limitations should be taken into consideration for future studies. We added the limitations of this study to the discussion,Line:344-350

Reference: Deng K, Liu Z, Su Y, Fan Y, Zhang Y, Wang F. Comparison of muscle fiber characteristics and meat quality between newborn and adult Haimen goats. Meat Sci. 2024;207:109361. doi:10.1016/j.meatsci.2023.109361

Statistical presentation has still issue.

For example, Table 1: interaction effect is not significant, but they use superscripts, for example, for FCR; for ADFI diet effect is significant, but no superscript has been used.

Response: Thanks. SIF supplementation did not exhibit any significant effect on ADFI (T×W; p = 0.511). In addition, SIF supplementation demonstrated a impact on FCR (T×W; p = 0.083) after 8-week treatment.

Table 3: Eye muscle area is not significant, but superscripts have been used.

Response: Thanks. We fixed the error.

Table 4, 5: Superscripts are not defined properly.

Response:Revised according to the comment. Annotations were given under the Tables for the significance of statistical analysis.

Figure 2 A: Were there no significant effect at the phylum level? It seems Firmicutes in control was greater.

Response: As substantiated in Fig. 2A and Supplemental Table S4. Goats supplemented with SIF2 had greater relative abundance of Bacteroidota than that of NC and SIF1 groups. Furthermore, SIF treatment led to lower relative abundance of Firmicutes and the SIF2 group had a more obvious effect than the SIF1 group. Line:288-292

Reviewer 2 Report

Comments and Suggestions for Authors

The authors have addressed most of the reviewer’s comments. Only a few issues remain.

Specific comments:

L66-68: Include a literature citation for this study.

L261-263: Rarefaction curves are useful, but the authors should calculate Good’s coverage, and include the range of values in the text.

L266: These two metrics are typically considered to assess diversity. Richness is typically measured using other metrics, such as Chao1 (see Gotelli and Chao, DOI: 10.1016/B978-0-12-384719-5.00424-X).

L352-354: Based on the data in Supplementary Table S5 (row 36), the authors should preface this sentence with a qualifier, “Although Papillobacter represented only 0.03-0.5% of 16S rRNA gene abundance when the ruminal samples were taken at 57d, …”.

L363-366: This conclusion is actually strengthened by the data.  According to Supplementary Table S5, row 37, Desulfovibrio represents <0.5% of the sequences, but other lactate utilizers (Megasphaera, Selenomonas) are even less abundant. This might be worth mentioning.

L376-379: It is worth adding here that additional testing at lower dosages may reveal beneficial effects as well.

Supplementary Table 1, L23: Insert formula for calculation, or appropriate literature citation.

Comments on the Quality of English Language

Only a few minor edits suggested:

L86: Change “have” to “had”.

L108: Change “individuals” to “people”, to allow the reader to realize immediately that the “individuals” were people, not goats!

L177: Insert “day” after “57th”. 

Supplementary Table 1, L19: Change “VA” to “Vitamin A”, and “VD3” to “Vitamin D3

Author Response

Specific comments:

L66-68: Include a literature citation for this

Response:Thank you for your comments. This section was rewritten according to the comment. (Line 68-71)

L261-263: Rarefaction curves are useful, but the authors should calculate Good’s coverage, and include the range of values in the text.

Response:We apologize for this unclear statement. Good’s coverage showed in Table S4. Good's coverage of each sample is greater than 0.995 and is not shown in the text.

L266: These two metrics are typically considered to assess diversity. Richness is typically measured using other metrics, such as Chao1 (see Gotelli and Chao, DOI: 10.1016/B978-0-12-384719-5.00424-X).

Response:Thanks for this comment. The sentence were rewritten according to comments and Chao1 showed in Table S4. (Line 274-275)

L352-354: Based on the data in Supplementary Table S5 (row 36), the authors should preface this sentence with a qualifier, “Although Papillobacter represented only 0.03-0.5% of 16S rRNA gene abundance when the ruminal samples were taken at 57d, …”.

Response:Thank you for your comments. This section was rewritten according to the comment. (Line 366-372)

L363-366: This conclusion is actually strengthened by the data.  According to Supplementary Table S5, row 37, Desulfovibrio represents <0.5% of the sequences, but other lactate utilizers (Megasphaera, Selenomonas) are even less abundant. This might be worth mentioning.

Response:Thank you for your comments. This section was rewritten according to the comment. (Line 381-386)

L376-379: It is worth adding here that additional testing at lower dosages may reveal beneficial effects as well.

Response:Thank you for your comments. This section was rewritten according to the comment. (Line 396-403)

Supplementary Table 1, L23: Insert formula for calculation, or appropriate literature citation.

Response:Thank you for your comments. OM were measured values and rewritten in Supplementary Table 1. ME was calculated according to NRC (2007).

Comments on the Quality of English Language

Only a few minor edits suggested:

L86: Change “have” to “had”.

L108: Change “individuals” to “people”, to allow the reader to realize immediately that the “individuals” were people, not goats!

L177: Insert “day” after “57th”. 

Supplementary Table 1, L19: Change “VA” to “Vitamin A”, and “VD3” to “Vitamin D3”

Response: Revised in Line 88, 110, 182 and Supplementary Table 1.

Additionally, the writing was revised throughout the manuscript. All the corrected sentences are marked in yellow in the manuscript.
